# The Effects of Exclusively Resistance Training-Based Supervised Programs in People with Depression: A Systematic Review and Meta-Analysis of Randomized Controlled Trials

**DOI:** 10.3390/ijerph17186715

**Published:** 2020-09-15

**Authors:** Lara Carneiro, José Afonso, Rodrigo Ramirez-Campillo, Eugenia Murawska-Ciałowciz, Adilson Marques, Filipe Manuel Clemente

**Affiliations:** 1Department of Sport and Physical Education, University Institute of Maia (ISMAI), Castêlo da Maia, 4475-690 Maia, Portugal; larafcarneiro@gmail.com; 2Research Centre in Sports Sciences, Health Sciences and Human Development, CIDESD, GERON Research Community, 5001-801 Vila Real, Portugal; 3Centre for Research, Education, Innovation and Intervention in Sport (CIFI2D), Faculty of Sport of the University of Porto (FADEUP), 4200-450 Porto, Portugal; jafonsovolei@hotmail.com; 4Human Performance Laboratory, Quality of Life and Wellness Research Group, Department of Physical Activity Sciences, Universidad de Los Lagos, Lord Cochrane, 1046 Osorno, Chile; r.ramirez@ulagos.cl; 5Department of Physiology and Biochemistry, University School of Physical Education, 51-612 Wroclaw, Poland; eugenia.murawska-cialowicz@awf.wroc.pl; 6Faculty of Human Kinetics, University of Lisboa, Cruz Quebrada, 1499-002 Lisboa, Portugal; amarques@fmh.ulisboa.pt; 7Escola Superior Desporto e Lazer, Instituto Politécnico de Viana do Castelo, Rua Escola Industrial e Comercial de Nun’Álvares, 4900-347 Viana do Castelo, Portugal; 8Instituto de Telecomunicações, Delegação da Covilhã, 1049-001 Lisboa, Portugal

**Keywords:** exercise, health, depressive symptoms, strength training, meta-analysis

## Abstract

The purpose of this study was to systematically review the effects of supervised resistance training (RT) programs in people diagnosed with depression or depressive symptoms. The following databases were used to search and retrieve the articles: Cochrane Library, EBSCO, PEDro, PubMed, Scopus and Web of Science. The search was conducted in late June 2020. Search protocol required the title to contain the words depression or depressive or dysthymia. Furthermore, the title, abstract or keywords had to contain the words or expressions: “randomized controlled trial”; and “strength training” or “resistance training” or “resisted training” or “weight training”. The screening provided 136 results. After the removal of duplicates, 70 records remained. Further screening of titles and abstracts resulted in the elimination of 57 papers. Therefore, 13 records were eligible for further scrutiny. Of the 13 records, nine were excluded, and the final sample consisted of four articles. Results were highly heterogeneous, with half of the studies showing positive effects of resistance training and half showing no effects. In two of the four combinations, the meta-analysis revealed significant benefits of RT in improving depressive symptoms (*p* ≤ 0.05). However, considering significant differences with moderate (Effect Size = 0.62) and small (ES = 0.53) effects, the heterogeneity was above 50%, thus suggesting a substantial level. To draw meaningful conclusions, future well-designed randomized controlled trials (RCTs) are needed that focus on understudied RT as a treatment for depression.

## 1. Introduction

Data from 2016 has revealed that depression is the leading cause of mental health-related disease burden globally, affecting an estimated 300 million people worldwide [1]. The WHO declared the SARS-CoV-2 (COVID-19) outbreak a global pandemic in March 2020, which has affected the lives of 71,429 people globally [2], and may potentially generate an increase in depressive states as a result of psychosocial stressors like life disruption, fear of illness, or fear of negative economic effects [3,4]. Therefore, it is important to understand how to approach depression and depressive symptoms in order to better prepare for an expected increase of people affected by it. The conventional treatment to treat depression is antidepressant medication. Besides pharmacotherapy, clinicians recommend cognitive behavior therapy and mindfulness-based therapy [5], as well as physical exercise [6]. Thus, treatment guidelines for mental illnesses from leading international organizations now recommend the integration of physical activity-based interventions as part of routine psychiatric care [7]. Exercise promises to be an efficacious treatment for people with depression. Indeed, several systematic reviews and meta-analyses have positively assessed the effects of exercise on depression [8,9,10,11,12,13,14].

Furthermore, persons with depression are at an increased risk of sedentary behavior [15], and exercise contributes to physical health in addition to mental health [6]. Specifically, evidence-based recommendations for the prescription of exercise for patients with major depressive disorders (MDDs) propose interventions of 2–3 sessions of supervised aerobic and/or aerobic and resistance training exercise of 45–60 min duration with moderate intensity per week [7], although it has been suggested that the volume of training may be more relevant than frequency [16]. To achieve optimal outcomes and decrease dropouts, the evidence also indicates that physiotherapists and qualified exercise professionals should lead and supervise physical exercise programs [17]. Even home-based programs, which may occur due to a plethora of reasons (e.g., distance, difficulties with transportation, costs), could benefit from being supervised through the utilization of new technological tools [18]. This could become more relevant if future scenarios similar to the COVID-19 situation are to be repeated and, therefore, force people to stay at home for prolonged periods of time.

The benefits of aerobic exercise (AE) for depression have been well documented; namely, it has been recognized that it promotes neurophysiological effects of which results may be similar to those observed after antidepressant drug treatments [19]. A meta-analysis of seven randomized controlled trials [20] showed that there was also good evidence that aerobic exercise over roughly 12 weeks improved cardiorespiratory fitness in people with depression. However, the effects of resistance training (RT) are less investigated, and the literature regarding RT as a stand-alone therapeutic intervention for MDD is limited [7]. Earlier studies that focused on RT for people with depression reported positive effects. However, these must be interpreted with caution due to the heterogeneity of the interventions and poor methodology by reporting several confounding variables that could interfere and introduce the potential for bias. As an example, Kim et al. [21] aimed to investigate the effects of 24 weeks of the RT program on depression in older women and their neurotransmitters. However, the RT intervention was composed of a 10 min warm-up consisting of walking and dynamic stretching, and then a 10 min cool-down with static stretching. Therefore, 20 min of the program, 40% of the session in part I, 33.3% in part II and 28.5% to 25% in part III, were devoted to activities that were not resistance training, and this may have contaminated the data.

To date, only one meta-analysis examined the efficacy of RT on depressive symptoms. Gordon and colleagues [22] analyzed 33 papers meeting their inclusion criteria. Their review concluded that RT significantly reduced depressive symptoms in adults regardless of health status, the total prescribed volume of RT, or significant improvements in strength. However, in the present systematic review and meta-analysis, which differs from the previous, more strict inclusion and exclusion criteria were defined. Namely, we excluded other comorbidities (e.g., Parkinson’s, Alzheimer’s, cancer and dementia) and only supervised exclusively resistance training-based interventions, with minimal warm-up activities outside the scope of the main exercise mode. Therefore, this systematic review and meta-analysis allows for a more comprehensive assessment of the potential benefits of RT for depression pathology. Hence, this systematic review with meta-analysis (SRMA) addresses one question: how effective is a stand-alone supervised intervention with RT at improving depression?

## 2. Materials and Methods 

This systematic review and meta-analysis followed the preferred reporting items for systematic reviews and meta-analyses (PRISMA) guidelines and the Cochrane Collaboration guidelines for the evaluation of risk of bias in randomized studies. The protocol was published in the International Platform of Registered Systematic Review and Meta-analysis Protocols with the number INPLASY202070004 and DOI 10.37766/inplasy2020.7.0004.

### 2.1. Eligibility Criteria

Articles were eligible if they were published or in press in a peer-reviewed journal, with full text in the English language. No limitations were placed regarding publication date, and articles in press were considered. The preferred reporting items for systematic reviews and meta-analyses (PRISMA) guidelines were adopted [23]. P.I.C.O.S. was established as follows: (i) Participants were humans explicitly diagnosed with any form of depression according to established criteria (e.g., the American Psychiatric Association’s [24] Diagnostic and Statistical Manual of Mental Disorders—DSM-5^®^ or previous versions, or the World Health Organization’s [25] International Classification of Diseases—ICD-11 or previous versions) or those with depressive symptoms above clinical threshold (cutoff values) determined by a validated screening measure (e.g., Beck Depression Inventory—BDI or BDI-II—by Beck et al. [26], the Hamilton Rating Scale for Depression—HAM-D—by Hamilton [27], the Geriatric Depression Scale—GDS—of Yesavage [28]), but without other major disease (e.g., Parkinson’s, Alzheimer’s, cancer, dementia); (ii) Only supervised exclusively resistance training-based interventions were considered, with minimal warm-up activities outside the scope of the main exercise mode; Comparators were control groups not performing any training protocol and/or supervised contrast groups also performing an alternative exercise program (i.e., yoga, stretching, aerobic exercise); Outcomes were any effects on performance, health and quality of life; Study design was limited to randomized controlled trials (RCTs).

### 2.2. Information Sources and Search

The following databases were used to search and retrieve the articles: Cochrane Library, EBSCO, PEDro, PubMed, Scopus and Web of Science. The search was conducted in late June 2020. Search protocol required the title to contain the words depression or depressive or dysthymia. Furthermore, the title, abstract or keywords had to contain the words or expressions: (i) “randomized controlled trial”; and (ii) “strength training” or “resistance training” or “resisted training” or “weight training”. No limitations were established for publication date, and in press articles were considered. For Cochrane Library, only trials were considered. For EBSCO, the title and abstract had to be searched separately, and, therefore, we chose to search (i) “randomized controlled trial”; and (ii) “strength training” or “resistance training” or “resisted training” or “weight training” without any limitations in the field. For PubMed, combined search only afforded the selection of title/abstract, not keywords. The same was valid for PEDro, besides only affording one field at a time. Therefore, multiple searches were needed in PEDro. In addition, the criterion of “clinical trial” was selected, therefore automatically excluding practice guidelines and systematic reviews. In Web of Science, the combination of title, abstract and keywords was termed “topic”.

### 2.3. Study Selection

The initial screening provided 136 results. After the removal of duplicates, 70 records remained. Screening of titles and abstracts resulted in the elimination of 57 papers. This was due to the following reasons: (i) non-scientific production (e.g., protocol registrations without actual research; book chapters, letters, replies); (ii) reviews (e.g., narrative or systematic reviews, meta-analyses); (iii) abstract-only records; (iv) original research where the exercise intervention did not apply resistance training; (v) original research where there was no group exclusively using resistance training; (vi) participants with other major health problems (e.g., cancer, Parkinson’s disease, dementia, end-stage renal disease); (vii) single group experiments; and (viii) papers unrelated to our topic. Therefore, 13 records were considered eligible for further scrutiny, and all were written in the English language.

Of the 13 records eligible for full-text analysis, nine were excluded. The paper by Kim, O’Sullivan and Shin [21] was excluded because the warm-up consisted of 10 min of walking and dynamic stretching, and the training session was followed by a 10 min warm-down consisting of static stretching. Therefore, 20 min of the program (representing 40% of the training in part I of the program, 33.3% in part II and 28.5% to 25% in part III) was devoted to activities that were not resistance training, and this may have contaminated the results. Similarly, the paper by Pereira et al. [29] included a 20 min warm-up consisting of walking and stretching, thereby meaning that 40% of the training session duration was not composed of resistance training. In a similar vein, Sims et al. [30] stated that there were warm-up and warm-downs besides resistance training, but there was no reporting of the duration and type of activities performed in these two stages. The paper of Teychenne et al. [31] was excluded because the strength and conditioning group had a mixture of resistance training and aerobic training. The papers of Ansai and Rebelatto [32], Chin et al. [33], Kekäläinen et al. [34], LeCheminant et al. [35], and Levinger et al. [36] were excluded because there was no diagnostic of depression and the average values of the utilized scales for evaluation of depression or depressive symptoms did not reach the cut-off value indicative of depressive symptoms. The final sample consisted of four articles: Krogh et al. [37], Moraes et al. [38], Sims et al. [39] and Singh et al. [40]. The process is synthesized in Figure 1. Two of the four articles included had more than one main outcome, resulting in the need to perform more than one meta-analysis (Figure 1).

### 2.4. Data Collection Process

JA and FMC conducted the initial search and the screening and exclusion process independently. LC later reviewed the entire process. After this stage, the entire process was compared step by step, and disagreements were discussed with all the authors of this manuscript until consensus was achieved.

### 2.5. Data Items

Study characteristics: (i) Sample size and general characteristics (e.g., age, sex/gender, physical activity habits); (ii) duration and characteristics of the intervention; (iii) adherence rates to the training protocol.

Primary outcomes: Mean change in depressive symptoms in the exercise group assessed by any validated scale, from baseline to post-intervention, in comparison with the mean change of the control and/or comparison groups. If an author reported the results of two outcome measures meeting our criteria (i.e., mean change/pre and post-test change in depressive symptoms according to two different measures), we used the primary outcome chosen by the author. If this was not clear, we used the HAM-D or the BDI to increase homogeneity in our results. These outcome measures were also prioritized since they were commonly used in the exercise and depression literature [9].

Secondary outcomes: (i) Physical (e.g., performance tests, body composition, perceived exertion); (ii) psychosocial (e.g., body image and appearance, reporting of positive or negative feelings, self-esteem, cognitive evaluations, memory and concentration tasks).

### 2.6. Risk of Bias in Individual Studies and Across Studies

The revised Cochrane risk-of-bias tool for randomized trials (RoB 2) was applied to evaluate the individual studies, considering its five dimensions: bias arising from the randomization process, bias due to deviations from intended interventions, bias due to missing outcome data, bias in the measurement of the outcome and bias in the selection of the reported results. JA and FMC completed the risk-of-bias evaluation independently. After completion of the first coding, the figures were compared, and all disagreements were discussed with all authors of the manuscript and reanalyzed until consensus was achieved.

### 2.7. Summary Measures, Synthesis of Results and Risk of Bias across Studies

The analysis and interpretation of results in this SRMA were only conducted if at least three study groups provided baseline and follow-up data for the same measure [41]. Means and standard deviations for a measure of pre-post RT interventions were converted to Hedges’ *g* effect size (ES). The inverse-variance random-effects model for meta-analyses was used because it allocated a proportionate weight to trials based on the size of their standard errors [42] and enabled analysis while accounting for heterogeneity across studies [43]. The ESs were presented alongside 95% confidence intervals (CIs) and interpreted using the following thresholds [44]: <0.2, trivial; 0.2–0.6, small; >0.6–1.2, moderate; >1.2–2.0, large; >2.0–4.0, very large; >4.0, extremely large. The analyses were carried out using the Comprehensive Meta-Analysis program (version 2; Biostat, Englewood, NJ, USA).

To estimate the degree of heterogeneity between the included studies, the percentage of total variation across the studies due to heterogeneity was used to calculate the *I*^2^ statistic [45]. Low, moderate and high levels of heterogeneity corresponded to *I*^2^ values of <25%, 25–75% and >75%, respectively [45].

Finally, the extended Egger’s test [46] was used to assess the risk of bias across the studies. In case of bias, a sensitivity analysis was conducted.

## 3. Results

### 3.1. Study Selection

As described in the methods, the initial search provided 136 articles, of which 70 remained after the removal of duplicates. Screening delivered 13 articles eligible for full-text analysis, after which nine papers were excluded with reasons that were previously explained. Four articles were included in the qualitative synthesis and meta-analysis. Table 1 presents the general data items for the individual studies.

### 3.2. Characteristics and Results of Individual Studies

The article by Krogh, Saltin, Gluud and Nordentoft [37] was an RCT with patients diagnosed with unipolar depression according to the ICD 10th revision. The sample of 165 patients (73.9% of which were women) was by far the largest among the included articles, potentially affording greater confidence in terms of the generalizability of the results. Furthermore, it was the longest trial, with 16 weeks of intervention. Three randomized groups of 55 patients each were formed: (i) RT; (ii) aerobic training; and (iii) relaxation, which had characteristics that approximated it to a true control group. Primary outcomes were assessed with the Hamilton scale, and there were no changes after the 16-week intervention. There were also no changes in secondary outcomes, such as quality of life and cognitive abilities. However, the resistance training group improved in 1 repetition maximum (1RM) testing, while the aerobic group improved in maximal oxygen uptake. Most importantly, there were reduced percentages of days absent from work in the RT group. The lack of broader and better results may, however, be due to very poor adherence rates to the training programs, with the top value of 56.2% being attributed to the RT group. Table 2 synthesizes the details concerning the RT protocols, Table 3 synthesizes the parallel group protocols, while Table 4 synthesizes the results for primary outcomes.

Moraes, Silveira, Oliveira, Matta Mello Portugal, Araújo, Vasques, Bergland, Santos, Engedal, Coutinho, Schuch, Laks and Deslandes [38] performed a 12-week, three-armed randomized intervention with 25 patients diagnosed with MDD according to DSM-4, and all taking medication, with the three groups (i) RT; (ii) aerobic training; and (iii) low-intensity exercise control. The small sample, especially the small number of patients per group, advises caution when generalizing the results. The adherence to the intervention was solid, with all patients engaging in ≥75% of the training sessions. After the intervention, both exercise groups outperformed the controls, showing significant reductions in both the Hamilton scale and the Beck Depression Inventory, suggesting that exercise improves the efficacy of exclusively pharmacological interventions. No secondary outcomes were assessed.

In their investigation, Sims, Galea, Taylor, Dodd, Jespersen, Joubert and Joubert [39] investigated 45 stroke survivor patients (stroke > 6 months before the investigation) diagnosed with depressive symptoms, but that were otherwise healthy at the beginning of the intervention, including in cardiovascular terms (see exclusion criteria in Table 1). The 10-week program compared an RT group with a non-exercise control group. Adherence was solid, with an average of 75% attendance rate to training sessions. The authors reported lower depression scores in the RT group after the 10-week intervention, but not after the 6-month follow-up. This should be interpreted as a normal detraining effect; after all, the patients exercised for 10 weeks, but later went through a detraining process lasting 14 weeks. However, even the short-term comparisons after the 10-week intervention should be taken with a grain of salt, as the differences in the Centre for Epidemiologic Studies for Depression Scale (CES-D) were already significant in baseline testing, with the exercise group presenting lower scores even before the intervention had started. There were no changes in secondary outcomes, such as quality of life, social support, self-esteem and other psychosocial assessments. Unsurprisingly, the RT group improved in 1RM strength testing.

Finally, Singh, Stavrinos, Scarbek, Galambos, Liber and Singh [40] selected 60 adults with some form of depression or depressive symptoms and randomized them into a high-intensity RT group, a low-intensity RT group and a control group. Adherence rates to the 8-week program were excellent, with over 95% average attendance rate to the training sessions. Primary outcomes were assessed using the Hamilton scale and GDS, and showed improvements in depressive symptoms for both experimental groups. Interestingly, this was the only study analyzing a dose-response relationship, and the improvements observed in the high-intensity group were much superior to those registered in the low-intensity group. However, the differences between the two experimental groups may have been exaggerated due to an excessive dissimilarity in exercise intensities. Firstly, one group worked with 80% 1RM, while the other was limited to 20% 1RM, which was already a very significant difference. Most importantly, though, while the 80% 1RM group was regularly re-tested to keep the loads adjusted at 80%, the 20% 1RM group kept the initial loads. Therefore, with adaptation to training, it is possible that by the later weeks, the low-intensity group was actually working with 10–15% 1RM. Additionally, this study showed that strength gains were associated with a decrease in depressive symptoms. The high-intensity group also experienced greater increases in quality of life.

### 3.3. Risk of Bias within Studies and across Studies

Risk of bias was assessed using Cochrane’s RoB 2 (see Table 5). Risk of bias arising from the randomization process was low for the articles by Krogh, Saltin, Gluud and Nordentoft [37] and Singh, Stavrinos, Scarbek, Galambos, Liber and Singh [40], but there were concerns with the other two papers [38,39]. Risk of bias due to deviations from intended interventions (effect of assignment to intervention) was low for all articles. Risk of bias in the effect of adhering to intervention was high in the paper of Krogh, Saltin, Gluud and Nordentoft [37], but low for the remaining articles. All articles had a low risk of bias due to missing outcome data. Risk of bias in measurement of the outcome was low for the paper of Krogh, Saltin, Gluud and Nordentoft [37], but there were some concerns with the other three papers. Finally, risk of bias was uniformly low for the selection of the reported results. Out of six dimensions, the article by Krogh, Saltin, Gluud and Nordentoft [37] had a high risk of bias in one dimension and low risk of bias in the remaining dimensions. The article by Singh, Stavrinos, Scarbek, Galambos, Liber and Singh [40] presented some concerns in one dimension, but low risk for the others. Finally, the articles by Moraes, Silveira, Oliveira, Matta Mello Portugal, Araújo, Vasques, Bergland, Santos, Engedal, Coutinho, Schuch, Laks and Deslandes [38] and Sims, Galea, Taylor, Dodd, Jespersen, Joubert and Joubert [39] raised some concerns in two dimensions, but had a low risk of bias in the others.

### 3.4. Meta-Analysis

Two of the articles [38,40] had two primary outcomes, which required different combinations resulting in four separate meta-analyses (Figure 2, Figure 3, Figure 4 and Figure 5). The relaxation group in the article by Krogh, Saltin, Gluud and Nordentoft [37] was considered a control group for practical purposes. First, the exercises were relaxation-based and with extremely low intensity, and therefore did not fit into recognizable categories of training programs. Furthermore, adherence rates to the sessions was <35%, meaning that this experimental group behaved very similarly to a true control group.

The following results represent the possible combinations for detecting the effects of RT in primary outcomes related to depressive symptoms. In the first combination, four randomized-controlled studies provided data for depressive symptoms, involving five experimental and four control groups (pooled *n* = 198). There was no significant effect of resistance training on depressive symptoms (ES = 0.39; 95% CI = −0.05 to 0.84; *p =* 0.082; *I*^2^ = 53.2%; Egger’s test *p* = 0.310; Figure 2). The relative weight of each study in the analysis ranged from 12.2% to 28.6%.

In the second combination, four randomized-controlled studies provided data for depressive symptoms involving five experimental and four control groups (pooled *n* = 198). There was a significant effect of resistance training on depressive symptoms (ES = 0.62; 95% CI = 0.03 to 1.20; *p =* 0.040; *I*^2^ = 71.9%; Egger’s test *p* = 0.155; Figure 3). The relative weight of each study in the analysis ranged from 13.7% to 25.3%. 

In the third combination, four randomized-controlled studies provided data for depressive symptoms, involving five experimental and four control groups (pooled *n* = 198). There was no significant effect of resistance training on depressive symptoms (ES = 0.48; 95% CI = −0.06 to 1.02; *p =* 0.081; *I*^2^ = 67.2%; Egger’s test *p* = 0.235; Figure 4). The relative weight of each study in the analysis ranged from 12.9% to 26.1%.

In the fourth combination, four randomized-controlled studies provided data for depressive symptoms, involving five experimental and four control groups (pooled *n* = 198). There was a significant effect of resistance training on depressive symptoms (ES = 0.53; 95% CI = 0.02 to 1.03; *p =* 0.040; *I*^2^ = 62.7%; Egger’s test *p* = 0.206; Figure 5). The relative weight of each study in the analysis ranged from 13.6% to 26.9%. 

## 4. Discussion

This meta-analysis is, to the best of our knowledge, the first to examine RCTs aimed at measuring the efficacy of exclusively RT supervised programs in people having depression. Although multimodal interventions are usually advised, understanding the role of each exercise modality is paramount to understanding the effects and necessity of such a training component. This is highly relevant health-wise, but also to assess whether time should be invested in a given training modality, or if better investments would be applied elsewhere. Therefore, the purpose of this study was to systematically review the effects of supervised resistance training programs in people diagnosed with depression or depressive symptoms. Upon retrieval of 136 papers (70 after the removal of duplicates), only four papers fulfilled the inclusion criteria, meaning the effects of resistance training on depressive symptoms is still not widely studied, against our expectations. The risk of bias was assessed with Cochrane’s RoB 2, with only a few selected concerns having arisen, but overall the four analyzed studies had a low risk of bias, and so we believe the reported results are trustworthy.

Primary outcomes focused on depressive symptoms. Here, results were highly heterogeneous, with half of the studies showing positive effects of resistance training and half showing no effects. In two of the four combinations (Figure 3, Figure 4 and Figure 5), the meta-analysis revealed significant benefits of RT in improving depressive symptoms (*p* < 0.05). Although considering significant differences with moderate (ES = 0.62) and small (ES = 0.53) effects, shown respectively in Figure 3 and Figure 5, the heterogeneity was above 50%, thus suggesting a substantial level [42]. In fact, in combinations 2 and 4 (Figure 3 and Figure 5), the experiment of Sims, Galea, Taylor, Dodd, Jespersen, Joubert and Joubert [39] was favorable to the control group, and the experiment of Moraes, Silveira, Oliveira, Matta Mello Portugal, Araújo, Vasques, Bergland, Santos, Engedal, Coutinho, Schuch, Laks and Deslandes [38] largely benefited the RT group. However, no experimental group experienced a worsening of symptoms as a result of resistance training.

The reason different combinations lead to different results appears to be a factor of analysis. It is important to note that several rating scales were used to assess the severity of depression in research and clinical settings. These measures were categorized as clinician-rated, such as the HAM-D, Montgomery Åsberg Depression Rating Scale (MADRS) [47] or Quick Inventory of Depression Symptomatology clinician rating [48], and self-reported scales, such as BDI and its revised version BDI-II, and Patient Health Questionnaire-9 [49]. In this study, as a primary outcome, HAM-D (17-item version), BDI, CES-D and GDS were used. Nevertheless, other studies utilized different assessment measures for their different interests. Choosing appropriate outcome measures is a fundamental component of any assessment. Therefore, selecting outcome measures could be considered as a compromise between factors. For example, depression measures should be selected based on the patient population. Indeed, depression occurs in children, adolescents, adults and the elderly. As a matter of example, GDS was specifically designed to screen and measure depression in geriatric patients. It contains 30 forced-choice “yes” or “no” questions, a format that is helpful for individuals with cognitive dysfunction [28]. 

Although various scales for rating depression severity have been developed to date, the HAM-D (17-item version) is the most regularly used clinician-rated scale in research and clinical settings. Originally published by Max Hamilton in 1960, the first version of the HAM-D, the HAM-D21, comprised 21 items [27]. Hamilton recommended, nonetheless, to use only the first 17 items of the HAM-D21 since the last four symptoms (i.e., diurnal variation, depersonalization, derealization, paranoid and obsessional/compulsive symptoms) were either not considered part of the illness, or they were relatively infrequent or not considered features related to depression severity [50]. On the other hand, the BDI is one of the most widely used self-rating scales. Thus, both the HAM-D and the BDI/BDI-II are frequently adopted as the primary outcome to assess depression severity in this scope [9]. In this line, it should be important to know what a given total score or a change score from baseline on one scale means in relation to the other scale.

This notion is supported by a review carried out by Furukawa et al. [51], which recently identified for the first time the corresponding scores of the HAM-D and the BDI (369 patients from seven trials) or the BDI-II (683 patients from another seven trials) using the equipercentile linking method. The results demonstrated that the HAM-D total scores of 10, 20 and 30 corresponded roughly with the BDI scores of 10, 27 and 42 or with the BDI-II scores of 13, 32 and 50. The HAM-D change scores of −20 and −10 corresponded with the BDI of −29 and −15 and with the BDI-II of −35 and −16. These results can help clinicians interpret the HAM-D or BDI scores of their patients in a more versatile way and also help clinicians and researchers assess such scores reported in the literature when scores on only one of these scales are provided. These dates were obtained from the RCTs of psychological and pharmacological treatments for major depressive disorders. However, despite being extensively used, the clinician-administered HAM-D [50] has been identified to present various psychometric problems, including lack of unidimensionality and poor ability to detect changes among persons with mild to moderate depressive symptoms [52]. These problems may be particularly relevant to studies investigating the antidepressant effects of exercise because exercise is a treatment that is particularly recommended for individuals with mild to moderate symptom severity. Consequently, the use of the HAM-D scale may not accurately reflect the magnitude of the antidepressant effect of exercise [53].

Another key point to address is related to a clear definition in the RCTs, which is the scale, aimed at measuring the main outcome. Therefore, it is crucial to provide clear information about it. Frequent omissions of key details, namely on the primary outcome and the scale used to measure the main outcome, may impair interpretability, replicability and synthesis of RCTs that could interfere with decision-making. 

In the first meta-analysis [22], which examined the efficacy of RT on depressive symptoms, there was wide variability in the scales used to measure depression, including Beck Depression Inventory (BDI); Brunel Mood Scale Questionnaire-Depression (BRUMS-D); Cardiac Depression Scale (CDS); Center for Epidemiologic Studies Depression Scale (CESD); Depression Adjective Checklist (DACL); Depression, Anxiety and Stress Scale (DASS-21); Geriatric Depression Scale (GDS); Hospital Anxiety and Depression Scale (HADS); Hamilton Rating Scale for Depression (HRSD); Major Depression Inventory (MDI); Mental Health Functioning Index–Depression (MHFI); Profile of Mood States–Depression (POMS-D), and Hopkins Symptom Checklist–Depression (SCL-90-D). To avoid misleading conclusions and to better understand the effects of RT on depression, it would be useful to have a more uniform use of measures across studies.

Another important topic is how to better frame the primary outcomes. It is very important to analyze adherence rates (i.e., compliance with training sessions). These were excellent in three of the four papers, but were very low in the study of Krogh, Saltin, Gluud and Nordentoft [37]. Therefore, despite being the longest study and having the greatest sample size, the very low adherence rates may have compromised the results of the interventions. Dropouts from exercise treatment represent a huge barrier that impedes individual benefit from an intervention, and appear to be a particular problem in persons with depression [54].

In this line, Stubbs, Vancampfort, Rosenbaum, Ward, Richards, Soundy, Veronese, Solmi and Schuch [17] conducted the first systematic review and meta-analysis which explored the incidence and predictors of dropout rates among adults with depression participating in exercise RCTs. These authors found that in MDD patients, higher baseline depressive symptoms predicted an increased dropout. Some strategies may serve as facilitators to reduce the impact of dropouts in those with MDD, namely sessions supervised by physiotherapists and exercise physiologists. Moreover, Schuch, Vancampfort, Richards, Rosenbaum, Ward and Stubbs [14] identified that, in people with depression who have higher social support, the likelihood of having symptom improvements in response to exercising increases. Additionally, incorporating a motivational component into exercise interventions for depression may be needed to decrease dropouts. Vancampfort et al. [55] found that autonomous motivation (i.e., acting out of choice and pleasure) was the key to adopting and maintaining physical activity behavior in patients with severe mental disorders defined as schizophrenia, bipolar disorder or MDD. Additionally, understanding that exercise is not a “one size fits all” intervention leading to immediate results is a key step to achieving progress [56].

At a certain point, [37] state that the American College of Sports Medicine’s guidelines suggest three weekly sessions, and lament the fact that their study could only provide two weekly sessions. However, as became apparent, even those two sessions had very low compliance rates. The classic study of Dunn, Trivedi, Kampert, Clark and Chambliss [16] assessed the dose-response relationship between frequency of activity (three or five times per week) and the amount of exercise based on energy expenditure (7 or 17.5 kcal/kg/week). The results demonstrated that the most important factor for decreasing depressive symptoms in people with mild to moderate depression seems to be the amount/dose of activity instead of frequency.

With these populations, a home-based supervised protocol could help. Indeed, home-based exercise programs may be an effective method to overcome barriers to exercise and increase exercise adherence. As a matter of example, Blumenthal, Babyak, Doraiswamy, Watkins, Hoffman, Barbour, Herman, Craighead, Brosse, Waugh, Hinderliter and Sherwood [19] randomized 202 individuals into four groups for 16 weeks of either home-based aerobic exercise, supervised group exercise, taking a placebo pill or taking sertraline. The authors concluded that individuals receiving active treatments tended to achieve higher remission rates than the placebo group: supervised exercise = 45%; home-based exercise = 40%; medication = 47%; placebo = 31% (*p* = 0.057). However, further research with larger sample sizes is needed to confirm adherence and efficacy for improving depressive symptoms. In any case, home-based exercise may be a good alternative to practicing physical activity, in particular for vulnerable people. 

Even in the other three studies, which had very good adherence to the programs, no study had more than two weekly training sessions. This may be a manifestly low frequency, perhaps insufficient to promote more expressive gains. It should also be highlighted that the study of Singh, Stavrinos, Scarbek, Galambos, Liber and Singh [40] presented the greater improvements in primary outcomes, despite being the shortest trial of only 8 weeks. The adherence rate of >95% presence in each training session can potentially explain this success. Therefore, training programs should be designed in a motivating manner to ensure high adherence rates, especially if only two weekly training sessions are performed. Motivation is a crucial predictor of success and a critical factor in supporting sustained exercise. 

Concerning secondary outcomes, only one study failed to assess physical outcomes [38]. This is unfortunate because the other three studies showed that even when depressive symptoms do not ameliorate, there are still benefits in terms of strength and cardiovascular response, which is positive in itself. Several reviews and studies have shown that people with severe mental illness, including people with MDD, have an excess mortality, being two or three times as high as those within the general population, and this excess mortality is mainly due to physical disease [57]. In this sense, and as an example, the rate of type II diabetes mellitus in individuals with MDD is around 8%, representing an increase of around 50% in comparison to the rate achieved for people without MDD [58,59]. A meta-analysis [20] revealed that individuals with depression could achieve clinically relevant improvements in cardiorespiratory fitness levels in response to exercise interventions. Furthermore, previous studies have found that exercise can improve physical and psychological domains of quality of life in individuals with severe mental illness [60], and specifically in people with depression [61]. Thus, exercise may act in these two directions, not only by improving symptoms of depression but also by improving the cardiovascular health and wellbeing of this vulnerable population.

Additionally, none of the four articles reported the rest times between exercises, although this a highly important and well-recognized variable to consider when prescribing RT. Indeed, when prescribed appropriately with other key prescriptive variables (i.e., volume and intensity), the amount of rest between sets could influence the efficiency, safety and effectiveness of an RT intervention [62]. Even if rest times were self-regulated in the studies, that should have been reported.

Relatively to psychosocial variables, effects on cognitive abilities and quality of life were scarce and heterogeneous. Moreover, only one study tested for a dose-response effect [40], with the high-intensity group having experienced not only the greatest improvements in depressive symptoms, but also in the quality of life. This raises the necessity of designing training programs that avoid excessively low intensities. However, despite the evidence which supports a dose-response relationship, people with depression typically have low cardiorespiratory fitness, so it seems to be idealistic to start with high volumes or intensities of exercise. Literature within this scope has shown that intensity is not critically significant for symptom management, and that exercise of even a light intensity can lead to short-term improvements in mood in this specific population [63]. For example, people with severe mental illness may face additional challenges towards exercise such as inexperience with intense physical efforts, associated fatigue and discomfort, increased risk of physical injuries, limited availability of physical activity facilities and specialized equipment, and costs associated with access to facilities or training [56]. Moreover, other barriers may include psychiatric medication side effects (e.g., sedation, fatigue, weight gain), lack of motivation and low self-confidence [64]. Lastly, recommendations for the prescription of exercise for patients with MDD should not be discarded. Small, incremental improvements can be obtained through real-life interventions aimed at improving the health of people with severe mental illness [7,65].

### Limitations

The existence of only four articles fulfilling inclusion criteria is daunting and reveals that this field has potentially not been adequately investigated. Moreover, the considerable heterogeneity of the studies (i.e., sample size and characteristics, experimental protocols, evaluated parameters, and so on) makes comparisons difficult and conclusions tentative. Also, only one study evaluated a dose-response effect [40]. Furthermore, it should be underlined that the four studies referred to samples from Brazil, Denmark and Australia. Therefore, out of the entire American continent, only South America is represented (and only by one country), while Europe also has a single representative. No data on the topic is available from Asia and Africa. This is troublesome, as the prevalence of depression may vary depending on a country’s human development index [66]. Patel et al. [67] carried out a systematic review and meta-analysis, which included 26 studies on developed countries. Approximately two-thirds of all studies and five out of six longitudinal studies showed a statistically significant positive relationship between income inequality and the risk of depression. Moreover, because societal context is paramount for understanding access, motivation and engagement with structured physical activity [68,69], cultural specificities may be important in understanding why, how and how often people have access to and engage with supervised exercise programs [70,71]. In the case of depression, country of origin, ethnicity and cultural differences are suspected of playing a major role in the patients’ symptoms and on the behaviors of their family aggregates [72].

## 5. Conclusions

In summary, the majority of interventions in this specific scope have been focused on aerobic exercise, and it is perhaps time to change the paradigm and invest in more research to assess the effects of RT in treating depression. Notwithstanding, to achieve this aim, it is essential to be rigorous and first assess the efficacy of exclusively RT supervised programs; otherwise, it may lead to misinterpretations. Additionally, it is crucial for those involved with the prescription of RT, namely exercise physiologists, to acquire an understanding of the program variables (e.g., loading and volume, exercise selection and order, rest periods, frequency) and the importance of their application [73]. Overall, to draw meaningful conclusions, future well-designed RCTs are needed that focus on understudied RT as a treatment for depression.

## Figures and Tables

**Figure 1 ijerph-17-06715-f001:**
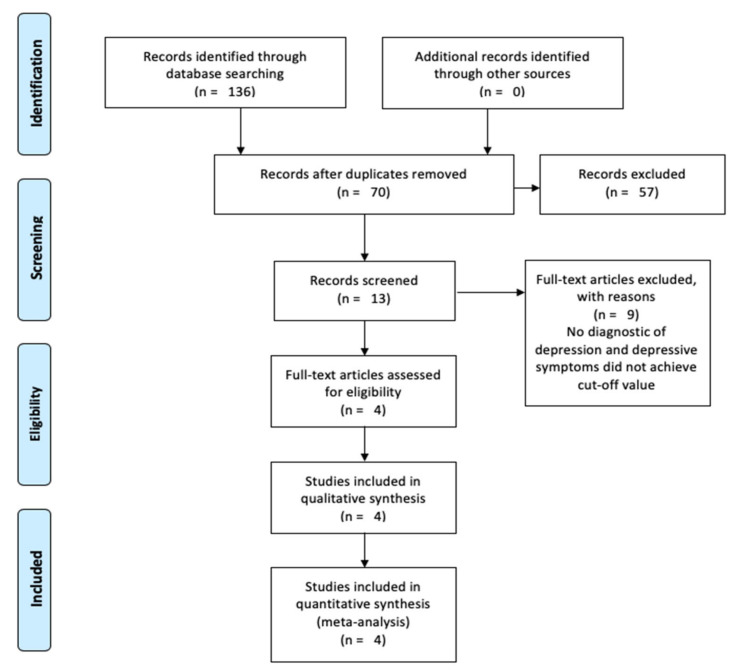
Flow diagram of the data collection process.

**Figure 2 ijerph-17-06715-f002:**
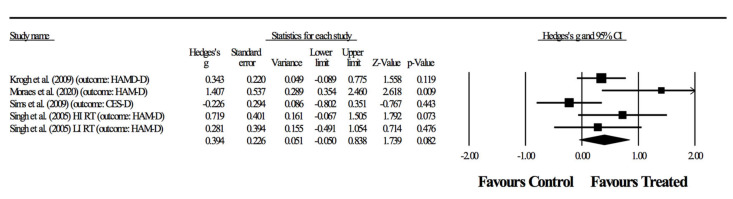
First combination in the meta-analysis. Forest plot of changes in depressive symptoms in participants diagnosed with depression (outcomes: three studies assessed through HAM-D, one with two groups, and another study using CES-D), after a supervised resistance training program compared to controls. Values shown are effect sizes (Hedges’ g), with 95% confidence intervals (CI). The size of the plotted squares reflects the statistical weight of the study.

**Figure 3 ijerph-17-06715-f003:**
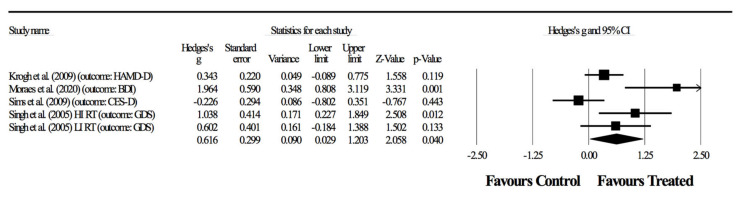
Second combination in the meta-analysis. Forest plot of changes in depressive symptoms (outcomes: one study with two groups assessed through GDS, one study using HAM-D, one using BDI, one using CES-D) in participants diagnosed with depression, after a supervised resistance training program compared to controls. Values shown are effect sizes (Hedges’ g), with 95% confidence intervals (CI). The size of the plotted squares reflects the statistical weight of the study.

**Figure 4 ijerph-17-06715-f004:**
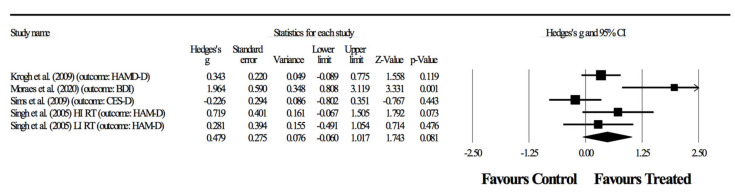
Third combination in the meta-analysis. Forest plot of changes in depressive symptoms in participants diagnosed with depression (outcomes: two studies assessed with HAM-D, one with two groups; another study using CES-D and one using BDI), after a supervised resistance training program compared to controls. Values shown are effect sizes (Hedges’ g), with 95% confidence intervals (CI). The size of the plotted squares reflects the statistical weight of the study.

**Figure 5 ijerph-17-06715-f005:**
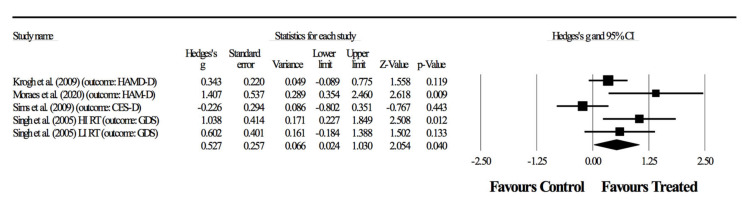
Fourth combination in the meta-analysis. Forest plot of changes in depressive symptoms in participants diagnosed with depression (outcomes: two studies assessed using HAM-D, one study with two groups using GDS and one using CES-D), after a supervised resistance training program compared to controls. Values shown are effect sizes (Hedges’s g), with 95% confidence intervals (CI). The size of the plotted squares reflects the statistical weight of the study.

**Table 1 ijerph-17-06715-t001:** General description.

Study	Population and Clinical Information	Groups	Adherence to Intervention	Primary Outcomes	Secondary Outcomes	Main Findings
Krogh et al. [37]	Randomized trial with patients diagnosed with unipolar depression according to ICD 10th revision. A total of 165 patients (122 women, 43 men) 18–55 years-old allocated to supervised training groups (3 to 10 participants per group). Exclusion criteria: suspected of psychotic symptoms, more than 1 h of sports per week, ongoing alcohol or substance abuse, considered at risk of suicide, poor Danish language skills, medical contraindications for exercise or patients having been on sickness leave for more than 24 consecutive months. A 4-month intervention with 12-month follow-up.	Resistance training (*n* = 55) and 46 at 12-month follow-up. Aerobic training (*n* = 55) and 46 at 12-month follow-up. Relaxation training (*n* = 55) and 37 at 12-month follow-up.	RT—Average 18.0 sessions out of 32 (56.2%). AT—Average 16.2 sessions out of 32 (50.6%). RT—Average 10.5 sessions out of 32 (32.8%).	17-item Hamilton Rating Scale for Depression (HAM-D_17_). Additional scales were used, but HAM-D_17_ was considered the main one by the authors.	Physical outcomes: 1RM in chest press, knee extension and leg press. Maximal oxygen uptake in cycle ergometer. Psychosocial outcomes Quality of life (WHO-5 Well-Being Index). Percentage of days absent from work during the last 10 working days (evaluated at four and 12 months) and other work-related information Cognitive abilities: Digit Span Test, Subtracting Serial Sevens, Trail Making Test, Digit Symbol Test, S phonemic fluency and verbal fluency test for animals, Buschke Test, Rey Complex Figure Test.	Exercise did not change primary outcomes, but RT reduced absences to work. RT group improved in 1RM chest press, while AT group improved in maximal oxygen uptake. No effect on cognitive abilities.
Moraes et al. [38]	Randomized trial with three exercise groups as adjunct treatments to pharmacotherapy (antidepressants and anxiolytics) for 25 persons diagnosed with major depressive disorder (MDD) according to DSM-IV, not engaged in physical exercise outside of the treatment setting. Patients were over 60 years old and sedentary for more than 3 months. Exclusion criteria: psychiatric comorbidities, score >18 points in HAM-D, score <24 on the Mini-Mental State Examination, cerebrovascular infarction, neurodegenerative disease, severe cardiovascular disease, illiteracy, poor mobility, balance disorders, and severe deficits in visual and/or auditory function. A 12-week intervention.	Resistance training (*n* = 9). Aerobic training (*n* = 9). Low intensity exercise control (*n* = 7).	All patients had a minimum of 75% attendance rate.	Hamilton Rating Scale for Depression (HAM-D_17_). Beck Depression Inventory (BDI), validated Portuguese translation.	None.	RT and AT groups showed significant reductions in depressive symptoms in both scales compared to controls, therefore improving upon the efficacy of pharmacological treatment only.
Sims et al. [39]	Randomized controlled trial with 45 stroke survivor patients (27 men and 18 women, 67.13 ± 15.23 years old) diagnosed with depressive symptoms using Prime-MD^®^ Patient Health Questionnaire-9 (PHQ-9) and confirmed by psychiatric assessment. A 10-week intervention with a follow-up at 6 months. Exclusion criteria: stroke <6 months before the study, inability to walk ≥20 m independently (with or without a gait assistive device), <18 years-old, PHQ-9 score <5, depression with psychotic features, alcohol or drug-related depression, schizophrenia, bipolar disorder, other psychiatric diagnoses, suicidal ideation, dementia, terminal disease, uncontrolled hypertension, unstable insulin-dependent diabetes and unstable angina.	Resistance training (*n* = 23). Waiting list comparison control (*n* = 22).	Average 75% adherence to the sessions.	Centre for Epidemiologic Studies for Depression Scale (CES-D).	Physical outcomes: 1RM for seated chest and leg press. Psychosocial outcomes: Assessment of Quality of Life Instrument (AQoL), Short Form-12 Health Survey Questionnaire (SF-12), Stroke Impact Scale (SIS) 3.0, Satisfaction with Life Scale (SWLS), Social Support Survey (SSS), Life Orientation Test-Revised (LOT-R), Self-Esteem Scale (Rosenberg, 1965), Recovery Locus of Control Scale (RLOC).	Authors report the RT group had lower depression scores after the intervention, but not at the 6-month follow-up. However, the RT group already had much lower depression scores at baseline. RT group improved significantly in strength, but ultimately there were no significant changes in CES-D from pre- to post or at follow-up.
Singh et al. [40]	Randomized controlled trial with 60 adults (33 women and 27 men, >60 years old) with major or minor depression or dysthymia, determined through DSM-IV, and who also had GDS score ≥14. Exclusion criteria: dementia, Folstein Mini-Mental State Examination score ≤23, medical contraindications for exercise, bipolar disorder, active psychosis, perceived suicidal tendencies, currently seeing a psychiatrist, prescribed antidepressant drugs in the previous 3 months, or participating in any exercise training more than twice a week.An 8-week intervention.	High intensity RT (*n* = 20). Eighteen completed the study. Low intensity RT (*n* = 20). Seventeen completed the study. Controls (*n* = 20). Nineteen completed the study.	There were six drop-outs. Of those who completed the study, adherence rates were >95%.	HAM-D_17_. GDS.	Physical outcomes: 1RM chest press, upright row, shoulder press, leg press, knee extension and knee flexion. Psychosocial outcomes: Eysenck Personality Questionnaire (EPQ) or EPQ-Revised, Wenger Social Support Network Instrument, Self-Efficacy Scale of Sherer, Multidimensional Health Locus of Control, Medical Outcomes Survey (Short Form 36), Pittsburgh Sleep Quality Index.	A 50% reduction in HAM-D in 61% of subjects of the high intensity RT group, 29% of the low intensity group and 21% of the controls. Strength gains were associated with a reduction in depressive symptoms. The high-intensity group had a superior decline in depressive symptoms and increases in quality of life.

RCT: randomized controlled trial; RM: repetition maximum. These studies were conducted in the following countries: Denmark (1 study), Brazil (1 study) and Australia (2 studies).

**Table 2 ijerph-17-06715-t002:** Type of resistance training protocol.

Study	W	S/w	Training Modality	WV (Min)	Exercises (*n*)	Sets (*n*)	Reps (*n*)	Intensity/Load (RM)	Rest Between Sets (Min)
Krogh et al. [37]	16	2	Circuit-training with machines, free weights and sandbags	90	10 (of which 6 were with machines)	2–3	1st phase: 12 2nd phase: 10 3rd phase: 8	1st: 50% 1RM 2nd: 75% 1RM 3rd: 75% 1RM	NR
Moraes et al. [38]	12	2	Machines	30	4	3	8–12	70% 1RM	NR
Sims et al. [39]	10	2	Machines	NR	6	3	8–10	80% 1RM	NR
Singh et al. [40]	8	3	High intensity and machines	60	6	3	8	80% 1RM	NR
	8	3	Low intensity and machines	60	6	3	8	20% 1RM	NR

W: weeks of intervention; S/w: session per week; WV: work volume; NR: not reported; RT: resistance training; RM: repetition maximum.

**Table 3 ijerph-17-06715-t003:** Type of parallel group training protocol.

Study	W	S/w	Training Modality	WV (Min)	Exercises (*n*)	Sets (*n*)	Reps (*n*)	Intensity/Load (RM)	Rest Between Exercises (Min)
Krogh et al. [37]	16	2	Aerobic training group: using machines, small carpets, trampoline, step bench, jump rope and Ski Fitter (Fitter International; Calgary, Alberta, Canada).	90	10 (of which 5 with machines)	2	1st phase: 2 min. Gradual increase to 3 min.	1st phase: 70% maximal HR. Gradual increase up to 89% maximal HR.	1st phase: 2 min. Gradual decrease to 1 min.
	16	2	Relaxation group: 20–30 min for exercises on mattresses or Bobath balls (Ledregomma; Udine, Italy) or back massage using Ball Stick Ball (Select; Glostrup, Denmark). 10–20 min of light balance exercises. 20–30 min of relaxation exercises with alternating muscle contraction and relaxation while lying down.	50–80	NR	NR	NR	<12 on the Borg scale.	NR
Moraes et al. [38]	12	2	Aerobic training group on stationary bikes or treadmills.	30	1	1	1	60% VO_2max_ or 70% HR_max_.	—
	12	2	Low-intensity control group: 5 min low-intensity walking or cycling ±15 min resistance training with minimum load±10 min stretching	30	8	1	1	Minimum possible	—

W: weeks of intervention; S/w: session per week; WV: work volume; NR: not reported; HR: heart rate.

**Table 4 ijerph-17-06715-t004:** Synthesis of results for primary outcomes.

Study	Group	N	Age	Pre (Mean)	Pre (SD)	Post (Mean)	Post (SD)	% Change (Pre-Post)	Follow-Up Test (Mean)	Follow-Up Test (SD)	% Change (Post Follow-Up)
Krogh et al. [37]	RT HAM-D_17_	46	41.9 ± 8.7	18.2	3.6	10.0	6.4	−45.1	11.0	7.1	10.0
	AT HAM-D_17_	46	38.1 ± 9.0	18.2	3.8	12.1	6.4	−33.5	11.9	6.5	−1.7
	Relaxation HAM-D_17_	37	36.7 ± 8.7	16.7	3.8	10.6	5.6	−36.5	10.0	5.6	−5.7
Moraes et al. [38]	RT HAM-D	9	72.9 ± 7.1	13.4	3.5	8.6	2.9	−35.8	—	—	—
	BDI			25.6	9.1	12.9	4.9	−49.6	—	—	—
	AT HAM-D	9	70.9 ± 5.9	14.3	2.82	7.4	2.1	−48.3	—	—	—
	BDI			19.7	6.44	12.8	3.6	−35.0	—	—	—
	LI-control HAM-D	7	69.3 ± 5.3	14.6	1.81	13.4	2.1	−8.2	—	—	—
	BDI			20.4	3.33	16.9	3.6	−17.2	—	—	—
Sims et al. [39]	RT CES-D	23	68.0 ± 14.8	15.4	7.49	15.1	8.5	−1.9	13.8	8.0	−8.6
	Control CES-D	22	66.3 ± 16.0	23.3	8.9	20.6	11.8	−11.6	22.7	11.2	10.2
Singh et al. [40]	HI RT HAM-D	18	69.0 ± 5.0	18.0	4.5	8.5	5.5	−52.8	—	—	—
	GDS			20.0	4.1	8.4	7.0	−58.0	—	—	—
	LI RT HAM-D	17	70.0 ± 7.0	19.5	5.3	12.4	6.3	−36.4	—	—	—
	GDS			22.0	4.3	13.3	7.0	−39.5	—	—	—
	Control HAM-D	19	69.0 ± 7.0	19.7	3.9	14.4	6.0	−26.9	—	—	—
	GDS			18.7	3.5	14.0	5.2	−25.1	—	—	—

RT: resistance training group; AT: aerobic training; PT: parallel training group; HI: high intensity; LI: low intensity; HAM-D: Hamilton Rating Scale for Depression; BDI: Beck Depression Inventory; GDS: Geriatric Depression Scale; CES-D: Centre for Epidemiologic Studies for Depression Scale.

**Table 5 ijerph-17-06715-t005:** Risk of bias (synthesized version *).

Cochrane RoB 2	Krogh et al. [37]	Moraes et al. [38]	Sims et al. [39]	Singh et al. [40]
1. Bias arising from the randomization process	Low	Some concerns	Some concerns	Low
2. Bias due to deviations from intended interventions (*effect of assignment to intervention*)	Low	Low	Low	Low
2. Bias due to deviations from intended interventions (*effect of adhering to intervention*)	High	Low	Low	Low
3. Bias due to missing outcome data	Low	Low	Low	Low
4. Bias in measurement of the outcome	Low	Some concerns	Some concerns	Some concerns
5. Bias in selection of the reported result	Low	Low	Low	Low

* Expanded version available upon request to FMC or JA.

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
