# Peer review of "The Effects of Exclusively Resistance Training-Based Supervised Programs in People with Depression: A Systematic Review and Meta-Analysis of Randomized Controlled Trials"

_ijerph, 2020, doi:10.3390/ijerph17186715_

Round 1

Reviewer 1 Report

It is an interesting paper allowing to understand the role of some types of physical excercise in the treatment of depression. The meta-analysis is used and thus the conclusions are based on more rigorous analyses than narrative or systematic review.   

I have few comments related to the text. These issues - in my view - should be addressed and clarified. These are isted below: 

  1. All the abbreviations should be explained when they appear first time in the text (e.g MDD in line 64);
  2. The inclusion criteria for the studies indicate ICD-11 (lines 106-108), but 3 out of 4 included studies used ICD-10 (then available as ICD-11 was published in 2018); the actual use of ICD-10 criteria in analysed studies is named in table 1 and line 216 - the Authors should correct their text to make is precise;
  3. Table 2 and 3 are not clear:  Why the same studies are described in 2 separate lines? It’s confusing and indicates more than 4 studies being analysed. I would suggest that all data concerning one study are given in one part of the table (without lines in-between)
  4. In line 250 the description of the study indicates that it included stroke survivors. It is not clear why such disease was accepted when other important confounding illnesses were not as indicated in the inclusion criteria? I think the Authors should comment on their decision.
  5. Different descriptions/titles  to figures 2-5 could be provided.  They should indicate which outcome measure was used. At present the titles are the same what might be confusing. Thus  either in the text or under the figure such information should be provided.
  6. Discussion – this section should avoid presentation of issues not addressed earlier, in introduction, like the text in lines 446-467.
  7. Comments on COVID, if any, should be included in the conclusions only as the analysed studies were conducted well before the pandemic.  Such comments can be related to the possible role of home-based exercise for MDD/other diseases especially when other forms are not possible.

Author Response

Reviewer 1: comments and responses:

  1. All the abbreviations should be explained when they appear first time in the text (e.g MDD in line 64)
    1. Such abbreviations are now explained in their first appearance. Lines were changes were made: 57, 92, 133-138, 256-257, 512-513.
  2. The inclusion criteria for the studies indicate ICD-11 (lines 106-108), but 3 out of 4 included studies used ICD-10 (then available as ICD-11 was published in 2018); the actual use of ICD-10 criteria in analysed studies is named in table 1 and line 216 - the Authors should correct their text to make is precise;
    1. This was a mistake on our part. It is now stated that previous versions of ICD were acceptable. The same for DSM.
  3. Table 2 and 3 are not clear: Why the same studies are described in 2 separate lines? It’s confusing and indicates more than 4 studies being analysed. I would suggest that all data concerning one study are given in one part of the table (without lines in-between)
    1. We agree and have now formatted tables 2 and 3 accordingly. Additionally, we have also proceeded in the same manner in table 4. Furthermore, we have noticed an inconsistency in decimal places in table 4 and are now uniformed. % changes were also lacking and were now added to table 4.
  4. In line 250 the description of the study indicates that it included stroke survivors. It is not clear why such disease was accepted when other important confounding illnesses were not as indicated in the inclusion criteria? I think the Authors should comment on their decision.
    1. This study was included because of its exclusion criteria, which were stated in table 1. All included patients had had their stroke more than 6 months before the study, were able to walk ≥20 meters independently and were otherwise healthy, including controlled tension values and no identifiable cardiac problems such as unstable angina. Therefore, they had had a stroke, but were currently relatively healthy and without limitations for engaging with the exercise program. This was now more explicitly stated in the manuscript (lines 504-506).
  5. Different descriptions/titles to figures 2-5 could be provided.  They should indicate which outcome measure was used. At present the titles are the same what might be confusing. Thus  either in the text or under the figure such information should be provided.
    1. We have added a more detailed description of the outcomes included in each figure. Furthermore, each figure now explicitly states which number of the combination it refers to (since there were multiple meta-analyses). Lines 704-706, 715-717, 735-737, 746-748.
  6. Discussion – this section should avoid presentation of issues not addressed earlier, in introduction, like the text in lines 446-467.
    1. We believe some of this information is relevant, but we also understand that it should have been mentioned in the introduction. Therefore, these issues are now addressed in the introduction, in lines 59-60, 62-66.
  7. Comments on COVID, if any, should be included in the conclusions only as the analysed studies were conducted well before the pandemic. Such comments can be related to the possible role of home-based exercise for MDD/other diseases especially when other forms are not possible.

In the introduction, the comments on COVID were reduced and, where maintained, their connection to the relevance of this article’s scope was better stated (e.g., lines 45-48). In the discussion, this issue was also toned down, with a few sentences being removed (e.g., near lines 874-876).

Reviewer 2 Report

I have the following comments for the authors to address:

1) Under the Introduction, the authors stated "The conventional treatment to treat depression is antidepressant medication, however, due to its side effects clinicians seek non-pharmacologic therapies, and one of these is physical exercise."  I think this statement is not correct. Psychiatrists suggest non-pharmacological therapies due to severity of depression and causes of depression. Not necessarily due to side effects. Furthermore psychotherapy such as cognitive behaviour therapy and mindfulness based therapy should be mentioned. I suggest to amend as follows:

The conventional treatment to treat depression is antidepressant medication.  Besides pharmacotherapy, clinicians recommend cognitive behavior therapy and mindfulness-based therapy (Ho et al 2020) and physical exercise [6].

Reference:

Ho CS, Chee CY, Ho RC. Mental Health Strategies to Combat the Psychological Impact of COVID-19 Beyond Paranoia and Panic. Ann Acad Med Singapore. 2020;49(3):155‐160.

2) Under the methodology, can the authors comment on whether they use random-effects model?

3) I suggest to move Table 5 to supplementary material as the paper is too long.

4) Can the authors describe what it means by work?

"only four works fulfilled the inclusion criteria". Is it papers?

5) Under Table 1, the authors should specify the countries that were included.

Under limitations, please state which continents were not included and the implications on the findings. Some cultural groups may be less open to exercise and this requires further discussion.

Author Response

Reviewer 2: comments and responses:

  1. Under the Introduction, the authors stated "The conventional treatment to treat depression is antidepressant medication, however, due to its side effects clinicians seek non-pharmacologic therapies, and one of these is physical exercise." I think this statement is not correct. Psychiatrists suggest non-pharmacological therapies due to severity of depression and causes of depression. Not necessarily due to side effects. Furthermore psychotherapy such as cognitive behaviour therapy and mindfulness based therapy should be mentioned. I suggest to amend as follows: <<The conventional treatment to treat depression is antidepressant medication.  Besides pharmacotherapy, clinicians recommend cognitive behavior therapy and mindfulness-based therapy (Ho et al 2020) and physical exercise [6]>>. Reference: Ho CS, Chee CY, Ho RC. Mental Health Strategies to Combat the Psychological Impact of COVID-19 Beyond Paranoia and Panic. Ann Acad Med Singapore. 2020;49(3):155
    1. We agree and have rephrased exactly as suggested by the reviewer. The suggested reference was also added. Now in lines 49-50.
  2. Under the methodology, can the authors comment on whether they use random-effects model?
    1. Yes, we did use random effects model. This had already been stated in the manuscript: << The inverse-variance random-effects model for meta-analyses was used because it allocates a proportionate weight to trials based on the size of their standard errors [42] and enables analysis while accounting for heterogeneity across studies [43]>>.
  3. I suggest to move Table 5 to supplementary material as the paper is too long.
    1. At the request of the editor, Table 5 was now drastically shortened and changed, and so we believe it should stay in the manuscript. However, we have also left a note under table 5, stating that the full version will be available upon request.
  4. Can the authors describe what it means by work? "only four works fulfilled the inclusion criteria". Is it papers?
    1. Yes, it is papers. We have now changed the term here and also in additional locations. Lines 169, 538, 540 and 760.
  5. Under Table 1, the authors should specify the countries that were included. Under limitations, please state which continents were not included and the implications on the findings. Some cultural groups may be less open to exercise and this requires further discussion.
    1. Under table 1, the countries of the participants in the analyzed papers are now provided (Denmark, Brazil and Australia). Furthermore, under limitations, we have now stated what continents were left out and potential implications. New text in lines 940-952.